# Polyphenols as Potential Protectors against Radiation-Induced Adverse Effects in Patients with Thoracic Cancer

**DOI:** 10.3390/cancers15092412

**Published:** 2023-04-22

**Authors:** Èlia Prades-Sagarra, Ala Yaromina, Ludwig J. Dubois

**Affiliations:** The M-Lab, Department of Precision Medicine, GROW—School for Oncology and Reproduction, Maastricht University, 6229 ER Maastricht, The Netherlands; e.pradessagarra@maastrichtuniversity.nl (È.P.-S.); ludwig.dubois@maastrichtuniversity.nl (L.J.D.)

**Keywords:** radiotherapy-induced adverse effects, polyphenols, radioprotection, normal tissue, thoracic cancers

## Abstract

**Simple Summary:**

Radiotherapy, commonly used to treat thoracic cancers, can induce adverse effects in the surrounding normal tissue, which become dose limiting factors and narrow the therapeutic window. Polyphenols, a type of natural plant compounds, have been proposed to radiosensitize the tumor and, at the same time, protect the normal tissue against radiotherapy-induced adverse effects. This review summarizes the current knowledge of the radioprotective effects of polyphenols, especially on normal thoracic tissues, and their underlying molecular mechanisms.

**Abstract:**

Radiotherapy is one of the standard treatment approaches used against thoracic cancers, occasionally combined with chemotherapy, immunotherapy and molecular targeted therapy. However, these cancers are often not highly sensitive to standard of care treatments, making the use of high dose radiotherapy necessary, which is linked with high rates of radiation-induced adverse effects in healthy tissues of the thorax. These tissues remain therefore dose-limiting factors in radiation oncology despite recent technological advances in treatment planning and delivery of irradiation. Polyphenols are metabolites found in plants that have been suggested to improve the therapeutic window by sensitizing the tumor to radiotherapy, while simultaneously protecting normal cells from therapy-induced damage by preventing DNA damage, as well as having anti-oxidant, anti-inflammatory or immunomodulatory properties. This review focuses on the radioprotective effect of polyphenols and the molecular mechanisms underlying these effects in the normal tissue, especially in the lung, heart and esophagus.

## 1. Introduction

More than half of cancer patients receive radiotherapy, either as monotherapy or in combination with other anti-cancer therapies. Many factors are involved in determining the final clinical outcome of radiotherapy, including the radiation type, dose, schedule and radiation delivery technique, as well as the final dose received by both tumor and normal tissue [1]. High-intensity radiotherapy is frequently needed due to cancer cell resistance to treatment, which increases the probability for radiation-induced toxicities in the normal tissues. In case of thoracic cancers, the most affected organs by radiation are the lungs, heart and esophagus [2,3,4]. These adverse effects are known as RILT, RIHD and radiation-induced esophagitis, respectively, and can be developed both at early stages, starting within approximately three weeks after the start of radiotherapy, and/or at late stages, with an onset of months to years after treatment [3,4,5]. In RILT, the most frequent toxicities are lung pneumonitis and fibrosis due to the high radiosensitivity of the lung parenchyma [6,7]. The incidence of pneumonitis after irradiation treatment in lung cancer patients stands at up to 50%, and pulmonary fibrosis is diagnosed in 70% to 80% of the patients [2,8]. Pneumonitis is developed at early stages and may be reversible, whereas pulmonary fibrosis is a late effect and considered irreversible due to the inexistence of an approved treatment. This leads to poorer prognosis and impaired quality of life [2]. RIHD includes a wide range of morbidities, including pericarditis, coronary artery disease, valvular heart disease, conduction abnormalities as well as cardiomyopathies [5,9]. RIHD is more frequent in long-term survivors, as it has a late onset and requires a long incubation period. Recent studies have shown that lung cancer patients receiving a heart radiation dose above 20 Gy showed a higher risk of RIHD and lower survival. More than 10% of lung cancer patients treated with thorax radiation suffered myocardial infarction, heart failure or cardiac death, with an onset of 18 months post-treatment [10,11]. In case of radiation-induced esophagitis, the most common early effects are dysphagia and odynophagia, which are described as difficulties to swallow or painful swallowing, respectively, and are developed in most patients within two months after treatment [4,12]. Other complications such as esophagus ulceration and perforation, as well as fibrosis at late stages, are less common, but more likely to occur when radiotherapy is combined with chemotherapy [4,13]. In fact, 1.3% of patients receiving radiotherapy as monotherapy develop severe esophagitis (>grade 3), while the incidence for patients receiving radiochemotherapy is up to 34% [12].

Radiotherapy adverse effects are therefore a major impediment to cancer treatment, as they become dose-limiting factors and narrow the therapeutic window [3], defined as the balance between TCP and NTCP [14]. The therapeutic ratio can be increased by either enhancing TCP, reducing NTCP or preferably a combination of both. In terms of TCP, efforts have been focused on increasing treatment response through, for example, the use of novel compounds, named radiosensitizers, which in combination with radiotherapy enhance the overall treatment effect, improving patient prognosis and survival [15,16]. On the other hand, chemical radioprotectors and improved dose conformity techniques are being investigated to decrease NTCP [3,17]. Furthermore, many studies are focused on widening the therapeutic window by increasing TCP as well as reducing NTCP. For example, fractionated irradiation, in which smaller radiation doses are delivered to the patient in multiple fractions, reduces NTCP as it causes less DNA damage in both tumor and normal tissues. As the normal tissue has a functional DNA repair system, it will be able to repair the damage in between the fractions and therefore decrease the normal tissue toxicities [18]. However, tumors have more aberrant DNA repair systems, which will result in insufficient repair by the next fraction, causing accumulation of sub-lethal DNA damage, which will likely become lethal at the end [18]. Another example is the use of modern high-precision radiation techniques such as IMRT to improve dose conformity, which permits the treatment of smaller and more accurate target volumes, reducing the dose delivered to the normal surrounding tissue [19,20]. Furthermore, state-of-the-art irradiation machines are capable of delivering a non-uniform dose beam, permitting the design of dose painting plans [20]. Dose painting, which consists of the application of different intensity radiation doses in the target, permits the delivery of a higher dose in the tumor, while sparing maximumly the normal tissue [21,22,23]. 

Pharmacological interventions using radioprotectors are being studied to prevent or reduce radiation-induced toxicities in the organs at risk within the irradiation field. Radiation leads to DNA damage as well as ROS formation, which in turn cause oxidation of DNA, lipids and proteins [24]. Radiation-induced DNA damage will induce the activation of many signaling pathways, which will cause inflammation and will modulate immune response [7]. A perfect radioprotector should therefore induce DNA repair mechanisms, have antioxidant properties as well as inflammatory and immunomodulatory effects, while having neither radioprotective effects on the tumor nor toxicity in other parts of the body [25]. Up to date, aminothiols and its derivatives, including amifostine, are the most investigated radioprotectors. Although these compounds have been shown to reduce and protect the normal tissue against radiation injuries in cancer patients, their clinical application is limited due to high toxicity [26,27]. So far, only amifostine has been FDA-approved, despite the controversy regarding its benefits and harms [26,28]. Evidence supports that amifostine could act as a ROS scavenger and an inducer of DNA damage repair mechanisms in normal, healthy cells [27,29]. However, its mechanism of action is complex and unclear, and its overall effect is being questioned due to the lack of complete specificity to the normal tissue and contradicting evidence regarding its radioprotective effects [25,26,27]. Its clinical use is also hampered due to its limited bioavailability and therefore difficulties in treatment administration and duration, toxicities in the healthy organs and elevated costs [26,30]. Further research is therefore required in order to identify novel radioprotectors which surpass these limitations.

Several plant-derived products such as polyphenols have gained interest within oncology due to their effects on both sides of the therapeutic window [31,32]. Polyphenols can sensitize tumors to radiation by interacting with several intracellular signaling pathways involved in tumor initiation and growth, such as the MAPK, the NF-κB and the Wnt/β-catenin pathway [33,34], activating tumor suppressor genes while downregulating oncogenes and pro-survival genes [31,35,36]. On the other hand, polyphenols’ protective effects have been studied in normal tissue including lung [37,38,39,40,41], breast [42,43], esophagus [44,45], skin [46,47] and intestine [48,49]. They behave as free radical scavengers, have anti-oxidant effects in DNA, lipids and proteins, and induce DNA repair mechanisms [24,31]. Previous research has shown that combined treatment of radiotherapy and polyphenols could increase the therapeutic window in thoracic cancer patients by radiosensitizing tumors as well as protecting the normal tissue, improving patient survival, outcome and quality of life [36,43,50]. This review focuses on summarizing the protective properties of polyphenols against radiation-induced thoracic organ toxicities (Figure 1).

## 2. Radioprotective Effects of Polyphenols in Thoracic Normal Tissues

Despite being a very large and heterogeneous family, polyphenols have been classified in four different categories—flavonoids (genistein, EGCG, silibinin, quercetin), phenolic acids (caffeic acid phenethyl ester, curcumin, thymol, zingerone), stilbenes (resveratrol) and lignans (secoisolariciresinol diglucoside) (Figure 2) [51]. In this review we summarize the radioprotective effects of each compound in thoracic organs in in vitro models, preclinical models and clinical trials.

### 2.1. Flavonoids

#### 2.1.1. Soy Isoflavones—Genistein

Soy isoflavones, such as genistein, are flavonoids (Table 1) mainly found in beans and legumes (including soybeans), and are considered estrogenic polyphenols as they mimic the action of estrogens on the estrogen receptors. In preclinical models of lung cancer, soy isoflavones have been shown to enhance the radiation cytotoxic effect by decreasing tumor growth and cell proliferation. In addition, soy isoflavones reduced radiation-related lung adverse effects including inflammation, hemorrhages and fibrosis [52,53]. Likewise, the administration of a genistein-enriched diet in both mice and rats has been shown to protect the animals against radiation-induced lung toxicities by reducing collagen deposition and consequently lung fibrosis [54,55]. Evidence supports that soy isoflavones, and specifically genistein, have an impact on the inflammatory responses that drive radiation-induced lung toxicities. The exact mechanism of action however is not clear yet. It was observed that genistein mitigates the radiation-induced immune response in mice by reducing leukocyte infiltrate in the lungs after daily soy isoflavones administration [56,57]. Soy isoflavones have been shown to affect blood vessel structure and expression of adhesion molecules by preventing nuclear activation of the transcription factor NF-κB [57,58]. Radiation-induced heart toxicities were mitigated upon soy isoflavones pre-treatment, improving heart structure and integrity, reducing damage in heart arteries as well as collagen deposition in arteries and myocardium, indicative for a reduction in radiation-induced heart fibrosis [59]. Soy isoflavones have also been proposed to reduce radiation-induced esophagitis in mouse models of lung cancer by reducing radiation-induced damage in different esophageal layers as well as immune cell infiltration [45]. Table 1 summarizes the preclinical studies on the radioprotective effects of soy isoflavones and their underlying molecular mechanisms.

#### 2.1.2. Epigallocatechin-3-Gallate (EGCG)

EGCG is one of the main bioactive components of green tea and has been reported to have cytotoxic and radiosensitizing effects by decreasing cell proliferation and inducing apoptosis in different cancers including lung and esophagus [24,60]. In normal tissues, EGCG has been suggested to protect against radiation-induced adverse effects (Table 2). In vivo, thorax irradiation rat models showed that EGCG improved animal survival as well as ameliorated radiotherapy-induced lung fibrosis due to its effects on oxidative stress and the Nrf2 signaling pathway [61]. 

Despite few in vitro and in vivo preclinical studies, several clinical trials have been performed in lung and esophagus cancer patients testing the radioprotective effects of EGCG. A phase I study in chemoradiotherapy-treated NSCLC patients revealed a lower degree of esophagitis as well as a reduced pain score upon EGCG supplementation [62]. This has been confirmed in two independent phase II clinical trials in lung cancer patients. EGCG not only decreased radiation-induced esophagitis but also other characteristic radiation injury symptoms such as pain, nausea and dysphagia [63,64]. In a prospective, three-arm phase II clinical trial comparing the effects of EGCG and placebo treatment in NSCLC patients receiving chemoradiotherapy, the degree of esophagitis was lower when EGCG was administered prophylactically, compared to a therapeutic treatment setup [50]. Similarly, a phase II trial in esophagus cancer patients treated with radiotherapy or chemoradiotherapy showed a reduced esophagitis score and pain upon EGCG treatment. Continuous EGCG treatment led to ameliorated esophagitis over time without affecting the anti-tumor therapy efficacy [44]. Phase III clinical studies are awaited. Table 2 summarizes the studies on the radioprotective effects of EGCG and its underlying molecular mechanisms.

**Table 2 cancers-15-02412-t002:** Radioprotective effects of epigallocatechin-3-gallate and its potential molecular mechanisms.

Model	Tissue	Treatment	Effect	Reference
Rat	Lung	- Thorax RT (22 Gy) - 25 mg/kg IP QD60	- ↓ MDA, ↑SOD activity in serum - ↑ Nrf2, HO-1, NQO-1 proteins in lung - ↓ serum IL-6, IL-10, TNF-α, TGF-β1 - ↓ RT-induced edema, hemorrhages in lung - ↓ collagen and myofibroblasts in lung - ↓ mortality	[61]
Clinical trial (phase I and II)	Esophagus	- Lung cancer patients - (Chemo)RT - EGCG oral solution (40–440 μM) once grade 1 esophagitis	- ↓ esophagitis grade - ↓ pain score - ↓ dysphagia	NCT02577393 [50,64]; NCT01481818 [62,63]
Clinical trial (phase I and II)	Esophagus	- Esophagus cancer patients - (Chemo)RT - EGCG oral solution (440 μM) once grade 1 esophagitis	- ↓ esophagitis grade - ↓ pain	NCT01481818 [44]

#### 2.1.3. Silibinin

Silibinin, one of the main active components of milk thistle (*Silybum marianum*), has been extensively studied for its anti-tumor effects in several types of carcinomas. Its ability to inhibit tumor growth has been demonstrated in both in vitro and in vivo models of lung cancer [65,66]. However, literature supporting its radioprotective effects is sparse (Table 3). An in vivo study using murine lung cancer models undergoing thorax irradiation demonstrated a reduction in radiation-induced lung toxicity and improved animal survival after oral administration of silibinin, as well as a reduced number of lung tumor nodules. Mice exhibited decreased inflammatory response in lungs as well as in BALF, and a mitigated lung fibrosis score [67]. Table 3 summarizes the studies on the radioprotective effects of silibinin and its underlying molecular mechanisms.

#### 2.1.4. Quercetin

Quercetin is a flavonoid present in fruits and vegetables, especially onions, berries and apples, as well as in green tea and red wine (Table 4). It has been suggested to have cytotoxic and radiosensitizing effects in many cancer types, including lung, by decreasing cell proliferation and inducing apoptosis in vitro [68,69]. In vivo models confirmed the radioprotective properties of quercetin against RILT as well as against other radiation-induced toxicities. The administration of quercetin (injected or inhaled) in murine and rat models of RILT showed a decrease in lung fibrosis severity and inflammatory infiltrate in BALF, plasma and lung tissue after total body irradiation [70,71,72]. One of the proposed mechanisms of action of quercetin is a decrease in the activity of NF-κB and MAPK pathways, supporting its anti-inflammatory effects [71,73]. The administration of quercetin has also been suggested to reduce radiation-induced oxidative stress and apoptosis in murine models [70,71]. 

No clinical trials have been performed so far testing the potential of quercetin to reduce radiation-induced thoracic toxicities. However, its safety and tolerance have been proven in patients with COPD [74]. Table 4 summarizes the preclinical studies on the radioprotective effects of quercetin and its underlying molecular mechanisms. 

### 2.2. Phenolic Acids

#### 2.2.1. Caffeic Acid Phenethyl Ester (CAPE)

CAPE (Table 5), obtained from propolis and considered one of its main bioactive components, has been proposed as a radiosensitizer for different cancer types. In case of lung cancer, CAPE has been proposed to induce cell death and decreased cell division in vitro [75,76]. CAPE is also considered a radioprotective agent due to its anti-inflammatory, anti-oxidant and immunomodulatory properties [77]. One of its mechanisms of action is the inhibition of the radiation-induced NF-κB pathway, which results in a reduced inflammatory response, i.e., reduction of pro-inflammatory cytokines combined with an upregulation of anti-inflammatory markers, in normal tissues and mitigated irradiation-induced pneumonitis [78]. Another mechanism of action of CAPE is the reduction of oxidative stress via its antioxidant properties, which has been shown in rat models of RILT. Animals showed decreased radiation-induced ROS as well as increased antioxidant enzymes [79]. CAPE has also been suggested to have radioprotective effects against radiation-induced heart toxicities. In vivo studies in rats showed that the administration of CAPE led to decreased oxidative stress as well as decreased pro-oxidant and increased anti-oxidant activity in heart tissue. Radiation-induced hyperlipidemia, which leads to oxidative stress, was also reduced. CAPE has also been suggested to mitigate radiation-induced heart toxicities by preventing the increase in serum cardiac enzymes [80]. Table 5 summarizes the studies on the radioprotective effects of CAPE and its underlying molecular mechanisms. 

#### 2.2.2. Curcumin

Curcumin, one of the major components of Indian turmeric (*Curcuma longa*), is a dietary polyphenol (Table 6) characterized for its antioxidant protective effects in normal tissue [31,81]. Furthermore, curcumin has also been proposed to have cytotoxic and radiosensitizing effects against different types of tumors, such as lung and cervical cancer by sensitizing cells to radiation as well as increasing apoptosis both in vitro and in vivo [31,81,82]. Curcumin has been suggested to have radioprotective effects in the normal tissue via scavenging ROS and preventing lipid peroxidation. Preliminary in vitro results showed that curcumin reduced the fraction of apoptotic lung cells and did not sensitize them to radiotherapy, which could be explained via its antioxidant properties [83,84]. In mouse lung fibroblasts, curcumin has been suggested to increase antioxidant enzymes activity as well as to prevent irradiation-induced ROS formation [85]. 

In mouse models of lung irradiation, both curcumin supplemented diet as well as liposome-delivered curcumin ameliorated RILT by reducing pneumonitis and lung fibrosis [81,85]. Curcumin has also been proposed to mitigate inflammatory responses via suppressing NF-κB activation as well as reducing pro-inflammatory cytokine levels [81]. Curcumin ameliorated RILT in rats independently of the administration route, including intragastric administration and curcumin-containing nanoparticle inhalation. Rats receiving curcumin treatment showed reduced radiation-induced pneumonitis as well as lung fibrotic tissue. Furthermore, curcumin exerted anti-inflammatory effects with a reduction in immune infiltration in the lung [83,86,87]. Inhaled curcumin also decreased oxidative stress via increasing antioxidant enzymes expression and reducing lipid peroxidation [83]. Curcumin has also been suggested to mitigate RIHD in rat models of thoracic irradiation by decreasing the inflammatory infiltrate as well as pro-cytokine levels in heart tissue [88]. Table 6 summarizes the studies on the radioprotective effects of curcumin and its underlying molecular mechanisms. 

#### 2.2.3. Thymol

Thymol (Table 7), obtained from thyme (*Thymus vulgaris*), has been suggested to sensitize certain tumor types to radiation, while protecting against treatment adverse effects in the normal tissue [24,89]. In vitro studies have confirmed its radioprotective effects in lung tissue as a result of its anti-oxidant properties. The administration of thymol prior to irradiation in hamster lung fibroblasts resulted in a reduction of radiation-induced apoptosis and necrosis [89,90], mediated through the prevention of radiation-induced DNA damage and mitochondrial membrane collapse [90]. In addition, thymol mitigated radiation-induced oxidative stress via a decrease in ROS levels and lipid peroxidation, as well as a prevention of the radiation-mediated reduction of antioxidant enzymes [89]. Table 7 summarizes the in vitro studies on the radioprotective effects of thymol and its underlying molecular mechanisms. So far, no preclinical studies have been performed to investigate the radioprotective effects of thymol in lung, heart and esophagus.

#### 2.2.4. Zingerone

Zingerone, one of the active compounds of ginger (*Zingiber officinale*), has also been suggested to have radioprotective effects (Table 8). However, literature supporting zingerone radioprotective effects in thoracic normal tissues is sparse. An in vitro study in hamster lung fibroblast cells proved zingerone to have anti-apoptotic and anti-oxidant properties. Pretreatment of zingerone improved cell survival after irradiation by inhibiting caspase-3 activation, reducing radiation-induced ROS levels and lipid peroxidation, as well as by increasing antioxidant enzymes levels [91]. 

The radioprotective effects of zingerone against RIHD have been evaluated in vivo in rat models. Intragastric administration of zingerone prior to radiation mitigated the radiation-induced changes in architecture of the myocardial muscle fibers. The levels of cardiac toxicity and apoptotic markers decreased compared to radiation alone. Zingerone also reduced the immune infiltration and inflammation in heart tissue. Lastly, zingerone increased antioxidant enzyme activity and reduced lipid peroxidation [92]. Table 8 summarizes the studies on the radioprotective effects of zingerone and its underlying molecular mechanisms.

### 2.3. Stillbenes

#### Resveratrol

Resveratrol, extracted from grapes and wine, berries and peanuts (Table 9), has been shown to have cytotoxic effects against tumor cells [31,93], while protection against radiation adverse effects in normal tissues due to its anti-oxidant properties [24,31]. In RILT mouse models, resveratrol has been proven to reduce the degree of radiation-induced pneumonitis and lung fibrosis, by decreasing the inflammatory response and immune cell infiltration in the lungs [94,95]. Although the radioprotective effects of resveratrol against RIHD are poorly understood, one study demonstrated that resveratrol is able to restore the heart metabolic profile of mice receiving radiation treatment. The levels of choline-containing compounds as well as lipids with unsaturated fatty chains, involved in the structure of the cellular membrane, were restored after resveratrol treatment [96]. Further confirmatory research however is warranted. Table 9 summarizes the preclinical studies on the radioprotective effects of resveratrol and its underlying molecular mechanisms. 

### 2.4. Lignans

#### Secoisolariciresinol Diglucoside (SDG)

SDG is one of the most common lignans and it is found in sesame, sunflower and pumpkin seeds, and flaxseeds (Table 10). SDG has been proposed to have cytotoxic effects against different cancer types, such as breast, colon and prostate [97,98,99]. In case of lung cancer, it has only been demonstrated that SDG does not have radioprotective effects in the tumor [100,101,102]. However, many studies support that SDG mitigates the radiation-induced adverse effects in the lungs due to its anti-inflammatory and anti-oxidant properties [100]. In vitro, SDG has been proposed to reduce radiation-induced DNA damage as well as oxidative stress by inducing antioxidant enzymes in normal lung cell lines [103]. In in vitro models of lung vasculature, SDG was observed to mitigate radiation-induced inflammation markers [104] as well as oxidative stress levels [103]. Ex vivo models of human precision cut lung slices also showed ameliorated radiation-induced adverse effects after proton irradiation and SDG treatment. SDG reduced radiation induced senescence, inflammation and oxidative stress [102]. 

In RILT mouse models, SDG supplemented diet prior to irradiation improved animal survival and welfare, together with reduced lung inflammation and fibrosis as well as oxidative stress [100,101,105]. Interestingly, the administration of SDG supplemented diet after several weeks post irradiation also mitigated the radiation-induced lung adverse effects in preclinical mouse models [106]. Table 10 summarizes the studies on the radioprotective effects of SDG and its underlying molecular mechanisms.

## 3. Discussion and Future Directions

In this review, we describe the effects of polyphenols against radiation-induced toxicity in lung, heart and esophagus. Polyphenols have been shown to have radioprotective effects also on other tissues including brain, breast, intestine, kidney, liver, prostate and skin (see Appendix A [42,43,107,108,109,110,111,112,113,114,115,116,117,118,119,120,121,122,123,124,125,126,127,128,129,130,131,132,133,134,135,136,137,138,139,140,141,142,143,144,145,146,147,148,149,150,151,152,153,154,155,156,157,158,159,160,161,162,163,164,165,166,167,168,169,170,171,172,173,174,175,176,177,178,179,180,181,182,183,184,185,186,187,188,189,190,191,192,193,194,195,196,197,198,199,200]). Furthermore, polyphenols have also been proposed to have radiosensitizing effects at the level of the tumor. This dual effect combined with its low toxicity is an interesting avenue in order to increase the therapeutic window, as these compounds could be able to increase tumor control and protect the normal tissue against the therapy-related adverse effects, increasing patient survival and quality of life. 

However, the administration of most polyphenols is hampered by their poor bioavailability due to their low solubility in water, which limits the delivery. For example, the average bioavailability of polyphenols when administered orally is 2 to 20% [201]. Furthermore, the content of polyphenols in certain types of food can also vary and therefore affect bioavailability when administered via diet, such as environmental factors, food processing and interaction with other molecules as well as human intestinal and general systemic factors (age, gender, previous pathologies, etc.) [202]. Another limitation frequently encountered when using polyphenols is their short half-life and therefore low tissue and plasma concentrations. Despite the fact that distribution of polyphenols varies depending on the administration route, most compounds are rapidly metabolized and excreted by the liver and kidneys into bile and urine, respectively [203]. Different strategies are being studied to overcome these limitations. Nanoencapsulation of polyphenols is becoming more common and studied in the research field to improve absorption and stability of these compounds. For example, many efforts have been made to encapsulate curcumin in lipoprotein particles, which resulted in improving its solubility while maintaining its radiosensitizing and radioprotective effects [81,84]. Another strategy is the development of polyphenol derivatives that have the same biological function with a better bioavailability and stability [204,205,206]. Derivatives of zingerone with higher water solubility were shown to have similar radioprotective effects as the natural compound, such as improved survival rate and prevented radiation-induced intestine toxicities in vivo [184]. 

Most studies testing the radioprotective effects of polyphenols in preclinical studies employed radiation techniques such as hemithorax, whole-thorax or whole-body irradiation. These experimental set-ups however do not represent current clinical practice, as here the irradiated area is larger and therefore the radiotherapy-adverse toxicities in the normal tissue might be more extensive [207]. This could lead to an underestimation of the radioprotective effects of polyphenols. Overall, further research should be performed to confirm the radioprotective effects of the discussed polyphenols and/or their novel derivatives with improved pharmacokinetic profiles both in preclinical studies, using state-of-the-art irradiation techniques reflecting the clinical situation which becomes possible with the development of small animal irradiators [208], and especially in clinical trials.

## 4. Conclusions

In conclusion, the use of polyphenols could be a promising strategy to increase the therapeutic window. Their dual radiosensitizing and radioprotective effect, combined with their low toxicity, suggests that they can increase tumor control, as well as protect the normal tissue against radiotherapy-induced toxicities, ultimately improving patient survival and quality of life. However, their poor bioavailability and short half-life may limit their effectiveness and administration. These limitations could be overcome via, for example, encapsulation of the compound or by developing derivatives of polyphenols with better solubility and equal biological effect. In addition, most preclinical studies performed do not mimic the current clinical practice, which could lead to an underestimation of their protective properties. Therefore, further research is needed to confirm their radio-protective effects in vivo as well as in clinical trials.

## Figures and Tables

**Figure 1 cancers-15-02412-f001:**
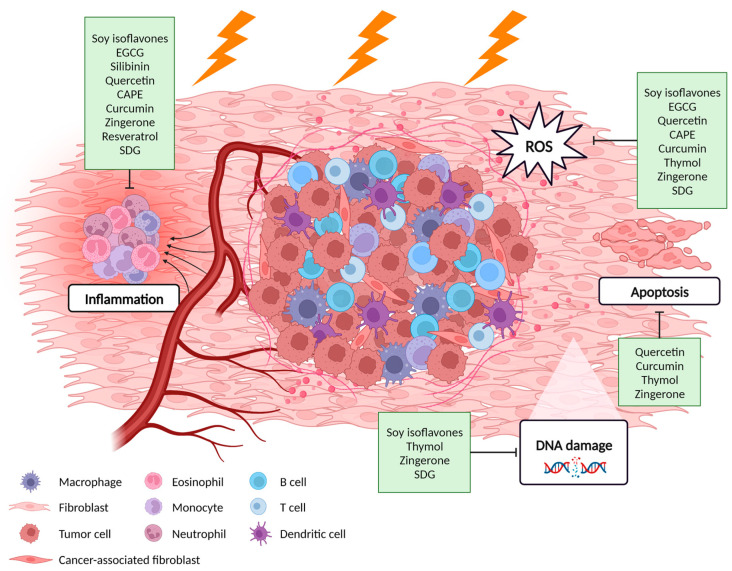
Potential molecular mechanisms of polyphenols against radiation-induced adverse effects in the thoracic normal tissue. Created with BioRender.com.

**Figure 2 cancers-15-02412-f002:**
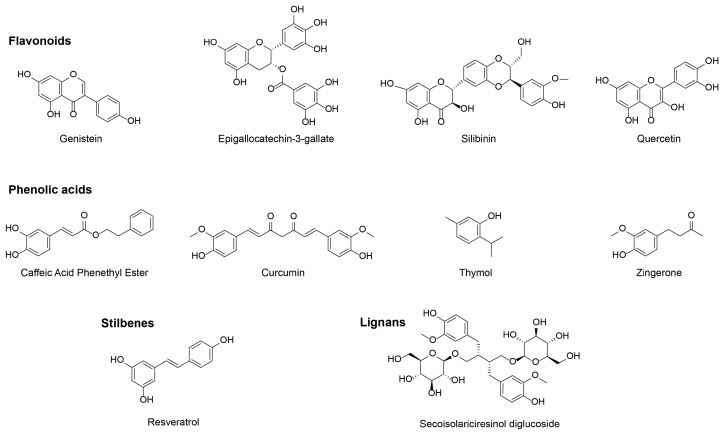
Chemical structure of the discussed polyphenols.

**Table 1 cancers-15-02412-t001:** Radioprotective effects of soy isoflavones and its potential molecular mechanisms.

Compound	Model	Tissue	Treatment	Effect	Reference
Soy isoflavones	Mouse	Lung	- Left lung RT (12 Gy) - Oral gavage - 1 mg/day QD3 prior to RT and QD-4W post RT daily	- ↓ lymphocyte, neutrophil infiltration - ↓ hemorrhage	[52]
Soy isoflavones	Mouse	Lung	- Left lung RT (10 Gy) - Oral gavage - 5 mg/day QD3 prior to RT - 5 mg/day QD5 after RT followed by 1 mg/day QD5-4W	- ↓ IL-6, IL-1β, IFN-γ, TNF-α - ↓ vasculature damage - ↓ alveolar septum thickness - ↓ interstitial collagen deposition in vessels and bronchioles	[53]
Soy isoflavones	Mouse	Lung	- Whole lung RT (10 Gy) - Oral gavage - 5 mg/day QD3 prior to RT - 5 mg/day QD10 after RT followed by 1 mg/day QD5-18W	- ↓ lymphocytes, macrophages and neutrophils in BALF and lung tissue - ↓ pro-inflammatory M1 macrophages in lung - ↑ anti-inflammatory M2 macrophages in lung	[56]
Soy isoflavones	Mouse	Lung (vasculature)	- Left lung RT (10 Gy) - Oral gavage - 12.5 mg/day QD3 prior to RT - 2.5 mg/day QD8 after RT	- ↓ leukocytes, MPO+ granulocytes in lung tissue - ↓ pro-inflammatory mediators of vascular endothelium (ICAM-1, VCAM)	[57]
Soy isoflavones	Mouse	Lung	- Whole lung RT (10 Gy) - Oral gavage - 5 mg/day QD3 prior to RT - 5 mg/day QD5 after RT followed by 1 mg/day QD5-4W	- ↓ NF-κB p65 protein levels in lung - ↓ RT-induced IL-6, IL-1β, IFN-γ, TNF-α - ↑ anti-inflammatory cytokine IL-15 - ↑ Arg-1 CD11b+ myeloid cells in lung	[58]
Soy isoflavones	Mouse	Heart	- Thorax RT (10 Gy) - Oral gavage - 5 mg/day QD3 prior to RT - 5 mg/day QD5 after RT followed by 1 mg/day QD5-16W	- ↓ artery wall aberrations and wall thinning - ↓ damage of artery muscle layer - ↓ collagen deposition in artery wall and myocardium	[59]
Soy isoflavones	Mouse	Esophagus	- Thorax RT (10 to 25 Gy) - Oral gavage - 5 mg/day QD3 prior to RT - 5 mg/day QD5 after RT followed by 1 mg/day QD5-16W	- ↓ leukocyte infiltration in esophagus tissue - ↓ esophageal muscle layer damage - ↓ collagen deposition in esophageal connective layers - ↑ proliferation in mucosal epithelium	[45]
Genistein	Mouse	Lung	- Whole lung RT - 9 fractions × 3.1 Gy (every 3–4 days) - Diet containing 750 mg/kg of genistein	- ↓ DNA damage (micronucleus formation) - ↓ macrophage infiltration - ↓ collagen deposition - Restored lung function (↓breathing rate vs. RT monotherapy)	[54]
Genistein	Rat	Lung	- Whole lung RT (18 Gy) - Diet containing 750 mg/kg of genistein	- ↓ DNA damage (micronucleus formation) - ↓ oxidative stress (8-OHdG staining) - ↓ IL-1β, IFN-γ, TNF-α - ↓ activated macrophage - ↓ collagen accumulation - ↑ survival - Restored lung function (↓breathing rate vs. RT monotherapy)	[55]

**Table 3 cancers-15-02412-t003:** Radioprotective effects of silibinin and its potential molecular mechanisms.

Model	Tissue	Treatment	Effect	Reference
Mouse	Lung	- Thorax RT (13 Gy) - 100 mg/kg PO, QD7 post-RT	- ↓ macrophages, lymphocytes in BALF - ↓ inflammatory infiltrate in lung - ↓ collagen deposition and fibrosis - ↓ thickening alveolar septum - ↓ number of tumor nodules	[67]

**Table 4 cancers-15-02412-t004:** Radioprotective effects of quercetin and its potential molecular mechanisms.

Model	Tissue	Treatment	Effect	Reference
Mouse	Lung	- Thorax RT (16 Gy) - Quercetin liposomes (mix quercetin/lecithin/cholesterol/PEG400 in 6:13:4:1) - 5 mg/kg IP, 2 h prior to RT and QD28 afterwards	- ↑ SOD, GSH peroxidase in lung - ↓ MDA levels in lung - ↓ lymphocytes, neutrophils in BALF - ↓ TNFα in plasma - ↓ TGF-β1 in plasma and lung - ↓ collagen deposition and fibrosis	[70]
Mouse	Lung	- Thorax RT (12 Gy) - Quercetin 10 mg/kg, IM, 1 h prior to RT (single dose)	- ↓ ROS, NO, MDA levels in lung - ↓ leukocytes, macrophages, neutrophils in BALF - ↓ IL-6, IL-1β, IL-18, TNF-α in BALF and serum - ↓ TGF-β1, Smad3 protein levels in lung - ↓ NF-κB, ICAM-1 protein levels in lung - ↓ apoptotic cells and caspase-3 expression (protein levels and activity) in lungs - ↓ collagen accumulation and fibrosis - ↓ RT-induced alveolar septum thickening	[71]
Mouse	Lung	- Thorax RT (10 Gy) - 4 mg/mL (100 μL), IV, 30 min before RT and QD2-6W afterwards	- ↓ lymphocyte, neutrophil infiltration in lung - ↓ RT-induced NF-κB pathway in lung - ↓ RT-induced MAPK pathway (↓ p38, SAPK/JNK, Erk1/2 expression) in lung - ↓ alveolar morphological changes (swelling, large fat droplets, necrosis) and RILT score	[73]
Rat	Lung	- Thorax RT (15 Gy) - 100 mg/kg quercetin, inhaled, QD7 prior to RT and QD-4M afterwards	- ↓ leukocyte in BALF - ↓ inflammatory infiltrate in lung - ↓ TGF-β1, IL-6 in lung - ↓ alveolar thickening, hemorrhages in lung	[72]

**Table 5 cancers-15-02412-t005:** Radioprotective effects of CAPE and its potential molecular mechanisms.

Model	Tissue	Treatment	Effect	Reference
In vitro	Human lung fibroblasts (WI-38)	- RT (9 Gy) - CAPE 6 μg/mL 1 h prior to RT	- ↓ ROS (H_2_O_2_), rescued GSH levels - ↓ RT-induced NF-κB protein levels - No radiosensitization	[78]
Mouse	Lung	- Thorax RT (10–20 Gy) - 10 mg/kg CAPE, IP, 30 min before RT and QD10 afterwards	- ↓ RT-induced NF-κB protein levels in lung - ↓ IL-6, IL-1α, IL-1β, TGF-β, TNF-α gene expression in lung - ↓ inflammatory infiltrate and pneumonitis in lung	
Rat	Lung	- Total body RT (0.8 Gy) - 50 μmol/kg CAPE, IP, 24 h prior to RT and QD3 afterwards	- ↑ CAT, SOD activity in lung - ↓ MDA levels in lung	[79]
Rat	Heart	- Total body RT (7 Gy) - 20 μmol/kg CAPE, IV, QD7 (starting 30 min post-RT)	- ↑ CAT, GSH peroxidase, SOD activity, ↓XO, ADA in heart - ↓ MDA, ↑total nitrate/nitrite levels in heart tissue - ↓ serum cardiac enzymes (AST, CPK, LDH) - ↓ RT-induced total cholesterol, LDL, triacylglycerids	[80]

**Table 6 cancers-15-02412-t006:** Radioprotective effects of curcumin and its potential molecular mechanisms.

Model	Tissue	Treatment	Effect	Reference
In vitro	Human bronchial cells (BEAS-2B)	- RT (15 Gy) - Curcumin nanoparticles (31.25 μM) for 24 h, prior to RT	- Rescued viability - ↓ RT-induced apoptosis	[83]
In vitro	Human lung fibroblasts (MRC-5)	- RT (0.05 Gy) - Curcumin nanolipoprotein disks (27 mg/mL)	- No radiosensitization in quiescent cells	[84]
In vitro	Mouse lung fibroblasts and pulmonary microvascular endothelial cells (primary)	- RT (2–6 Gy) - Curcumin (0–100 μM), 4 h prior to irradiation	- ↑ HO-1 activity - ↓ ROS levels - No radiosensitization	[85]
Mouse	Lung	- Right thorax RT (14 Gy) - Liposomes containing curcumin (10 mg/kg), IV, QD7 starting one day post RT	- ↓ IL-6, IL-8, TNF-α, TGF-β levels in plasma - ↓ collagen deposition and fibrosis - Delayed radiopneumonitis onset	[81]
Mouse	Lung	- Thorax RT (13.5 Gy) - Curcumin supplemented diet (1 to 5% curcumin w/w) for 7 to 14 days	- ↓ TNF-α in BALF - ↓ collagen and fibrosis - ↑ long term survival	[85]
Rat	Lung	- Thorax RT (15 Gy) - Inhaled curcumin nanoparticles (2.5 mg/kg), 5 h prior to RT	- ↑ SOD activity, ↓MDA content in lung - ↓ IL-6, IL-1β, TNF-α, TGF-β1 in lung - ↓ immune cells infiltrate, hemorrhaging in lung - ↓ collagen and fibrosis in lung - ↓ RT-induced hematopoietic damage (↑ leukocyte counts) - ↑ body weight	[83]
Rat	Lung	- Thorax RT (15 Gy) - Oral gavage - Curcumin (150 mg/kg), QD4 prior and QD6 post-RT	- ↓ IL-4 levels in lung - ↓ IL4ra1, DUOX1, DUOX2 gene expression in lung - ↓ macrophage, lymphocyte infiltrate in lung - ↓ collagen and fibrosis - ↓ alveolar and vascular thickness	[86]
Rat	Lung	- Thorax RT (18 Gy) - Oral gavage - Curcumin (200 mg/kg), QD7 prior and QD5-9W post-RT	- ↓ TNF-α, TNFR1, TGF-β1 protein levels in lung - ↓ NF-κB activation (protein levels, translocation to nucleus and acetylation) in lung - ↓ COX-2 protein levels in lung - ↓ macrophage infiltration - ↓ alveolar thickening, edema in lung - ↓ collagen deposition	[87]
Rat	Heart	- Thorax RT (15 Gy) - Curcumin 150 mg/kg daily, QD7 (starting 1 day prior to RT)	- ↓ IL-4 expression (protein and gene) in heart - ↓ DUOX1, DUOX2 gene expression in heart - ↓ lymphocyte, macrophage in heart	[88]

**Table 7 cancers-15-02412-t007:** Radioprotective effects of thymol and its potential molecular mechanisms.

Model	Tissue	Treatment	Effect	Reference
In vitro	Chinese hamster lung fibroblast cells (V79)	- RT - Thymol, prior to RT (0–120 μg/mL)	- ↓ ROS levels - ↓ lipid peroxidation (TBARS) - ↑ CAT, GSH, GST, SOD activity - ↓ apoptotic fraction - ↑ cell viability and radioprotective effect	[89]
In vitro	Chinese hamster lung fibroblast cells (V79)	- RT - Thymol, prior to RT (0–120 μg/mL)	- ↓ DNA damage and micronuclei - ↓ apoptotic, necrotic fraction - ↓ RT-induced mitochondrial membrane depolarization	[90]

**Table 8 cancers-15-02412-t008:** Radioprotective effects of zingerone and its potential molecular mechanisms.

Model	Tissue	Treatment	Effect	Reference
In vitro	Chinese hamster lung fibroblast cells (V79)	- RT - Zingerone, prior to RT (0–100 μg/mL)	- ↓ DNA damage and micronuclei - ↓ ROS levels - ↑ CAT, GSH, GST, SOD activity - ↓ lipid peroxidation (TBARS) - ↑ cell viability and radioprotection - ↓ apoptotic fraction - ↑ Bcl-2,↓Bax, Caspase-3 - ↓ RT-induced mitochondrial membrane depolarization	[91]
Rat	Heart	- Total body RT (6 Gy), at last day of zingerone treatment - Oral gavage - Zingerone (25 mg/kg), QD7-3W	- ↑ GSH content, CAT activity in heart - ↓ MDA levels in heart - ↓ serum TNF-α - ↓ COX-2 protein levels in heart - ↓ myeloperoxidase in heart - ↓ Caspase-3 gene expression in heart - ↑ ETC complexes activity - ↓ serum cardiotoxicity markers (BNP, cTNT levels; CK-MB, LDH activity) - ↓ RT-induced myocardial damage (hemorrhages, fat vacuoles)	[92]

**Table 9 cancers-15-02412-t009:** Radioprotective effects of resveratrol and its potential molecular mechanisms.

Model	Tissue	Treatment	Effect	Reference
Mouse	Lung	- Lung RT (18 Gy) - Oral gavage - Resveratrol 100 mg/kg, 24 h prior to RT and QD5-2W after RT	- ↓ neutrophil infiltration in lung - ↓ RT-induced morphological damage (alveolar thickening, congestion, edema, hemorrhaging) - ↓ collagen deposition and fibrosis	[94]
Mouse	Lung	- Thorax RT (18 Gy) - Oral gavage - Resveratrol 100 mg/kg, QD2 prior to RT and QD7 after RT	- ↓ lymphocyte, macrophage, mast cells infiltration in lung - ↓ RT-induced vascular and alveolar thickness - ↓ collagen and fibrosis	[95]
Mouse	Heart	- Heart RT (2 Gy) after 4 weeks of resveratrol treatment - Resveratrol in drinking water (5–25 mg/kg) for 6 weeks	- ↑ choline-containing metabolites - ↑ unsaturated lipids	[96]

**Table 10 cancers-15-02412-t010:** Radioprotective effects of SDG and its potential molecular mechanisms.

Model	Tissue	Treatment	Effect	Reference
In vitro	Murine lung epithelial cells, fibroblasts and endothelial cells	- RT (2–8 Gy) - SDG (0–50 μM), 0–6 h prior to RT	- ↓ DNA damage (γH2AX) - ↑ Gstm1, HO-1, NQO1 gene expression - ↑ HO-1, NQO1 protein levels - Restored proliferation capacity	[103]
Ex vivo	Human precision cut lung slices	- Proton irradiation (4 Gy) - Synthetic SDG (0, 50, 100 μM), 4 h prior to RT	- ↓ ROS levels - ↑ HMOX1, NQO1 gene expression - ↓ COX-2, IL-1β, IL-6 gene expression - ↓ ICAM-1, IL-1β protein - ↓ radiation-induced senescence and senescence genes (TP53, CDKN2A) - ↑ phosphorylated pRb levels - ↑ cell proliferation gene expression (CDK2, CDK4, CDK6)	[102]
In vitro	Pulmonary microvascular endothelial cells	- High LET/Low LET proton and gamma RT (0.25–1 Gy) - Synthetic SDG (0–100 μM), 30 min prior to RT	- ↓ ICAM-1 expression (protein) - ↓ NLRP3 inflammasome activation	[104]
In vitro	Murine pulmonary microvascular cells	- RT (2 Gy) - SDG (0–50 μM)	- ↓ ROS levels	[101]
Mouse	Lung	- Thorax RT (13.5 Gy) - 10% SDG supplemented diet, ad libitum starting 3 weeks prior to RT	- ↓ MDA levels in lung - ↓ Bax, p21, TGF-β1 gene expression in lung - ↓ neutrophils, macrophage in lung - ↓ collagen deposition and fibrosis	
Mouse	Lung	- Thorax RT (13.5 Gy) - 10% or 20% SDG supplemented diet starting 24, 48 or 72 h post-RT, ad libitum	- ↓ oxidative stress (TBARS) in BALF - ↓ MDA levels in BALF - ↓ TGF-β1 in lungs - ↓ FGF-β, IL-12, VEGF in BALF - ↓ leukocytes in BALF - ↓ collagen deposition and fibrosis in lung - ↑ survival - Restored cardiac and lung function (blood oxygenation levels)	[105]
Mouse	Lung	- Thorax RT (13.5 Gy) - 10% or 20% SDG supplemented diet, ad libitum starting 3 weeks prior to RT	- ↑ HO-1, NQO1, Nrf2 gene expression in lung - ↑ HO-1 protein levels in lung - ↓ leukocytes in BALF - ↓ RT-induced morphological damage (alveolar thickening, edema) - ↓ collagen deposition and fibrosis - ↑ survival, restored BW - Restored cardiac and lung function (blood oxygenation levels)	[100]
Mouse	Lung	- Thorax RT (13.5 Gy) - 10% SDG supplemented diet, ad libitum 3 weeks before RT or 0/2/4/6 weeks post-RT	- ↓ MDA levels in lung - ↓ IL-6, IL-12, IL-14, VEGF in BALF - ↓ leukocyte in BAL - ↓ collagen deposition and fibrosis - ↑ survival, restored BW - ↓ RT-induced morphological damage (alveolar thickening, edema) - Restored cardiac and lung function (blood oxygenation levels)	[106]

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
