# Peer review of "Polyphenols as Potential Protectors against Radiation-Induced Adverse Effects in Patients with Thoracic Cancer"

_cancers, 2023, doi:10.3390/cancers15092412_

Round 1
Reviewer 1 Report
This review is very well written and provides a throughout overview of the body of evidence for the use of polyphenols to protect tissues during RT.
The tables are well structured and informative.
The conclusions are supported by the data presented. The authors could have expanded perhaps on the delivery issues and reported on the nanosolutions that are being tested since the difficulty in delivery is identified as the key barrier to clinical translation.
Author Response
Thank you very much for your effort reviewing our manuscript, and your positive comments. As suggested, we have extended the discussion on the delivery of polyphenols and the different strategies to improve their pharmacokinetics (highlighted changes on lines 387-394).
Reviewer 2 Report
In Figure 1, there is a need to define all different types of cells shown.
Also, in Figure 1, is it correct that authors mention inhibition of apoptosis by several polyphenols. Is this correct and what’s the perspective ?
Introduction is a little too long and authors can think about reducing it to provide an overview and background information first before truncating a large part of the write-up to focus on a new section 2 on polyphenols and the mechanisms discussed.
Authors have loosely grouped all 3 cancers as ‘thoracic’ cancers which is not correct. This can be confusing for beginners. This needs to be clarified and edited, as appropriate, through out the manuscript.
In continuation of above concern, this reviewer believes that breast cancer write-up is inferior to that for lung cancer. Combined with the above concern, it would make more sense to just focus on lung cancer.
Section 3 is definitely out of scope because between this and the earlier sections, authors cover every cancer !! Also, given the amount of literature that can potentially be covered in section 3, the write-up does not do justice.
Abstract is too vague and does not provide a snap shot of contents, for example, the polyphenols discussed, major mechanisms etc.
Author Response
- In Figure 1, there is a need to define all different types of cells shown.
A legend has been included for the different types of cells shown in Figure 1.
- Also, in Figure 1, is it correct that authors mention inhibition of apoptosis by several polyphenols. Is this correct and what’s the perspective?
Polyphenols have been suggested to decrease radiation-induced apoptosis in normal tissue. See for example line 223 (quercetin, ref. 73), 261 (curcumin, ref. 76), 293 (thymol, ref. 96). These effects are reflected in Figure 1 as inhibition of apoptosis in normal tissue.
It should be taken into consideration that some polyphenols have been shown to induce apoptosis in tumor cells, for example line 165 (EGCG, ref. 23), 215 (quercetin, ref. 70, 71), 258 (curcumin, ref. 84).
- Introduction is a little too long and authors can think about reducing it to provide an overview and background information first before truncating a large part of the write-up to focus on a new section 2 on polyphenols and the mechanisms discussed.
We believe that all information mentioned in the introduction section is essential. We decided to make the introduction more compact by removing nonessential sentences rather than truncating a large part of the introduction.
- Authors have loosely grouped all 3 cancers as ‘thoracic’ cancers which is not correct. This can be confusing for beginners. This needs to be clarified and edited, as appropriate, through out the manuscript.
We have discussed with a radiation oncologist if the term “thoracic cancers” is correct. He confirmed that is correct to use “thoracic cancers” for all cancers in the thoracic area, including lung and esophagus. Therefore, we would like to use this term in this manuscript. However, we make it clear in the abstract and introduction, what we mean by thoracic cancers.
- In continuation of above concern, this reviewer believes that breast cancer write-up is inferior to that for lung cancer. Combined with the above concern, it would make more sense to just focus on lung cancer.
We agree with the reviewer that the data shown for breast cancer is inferior compared for example to lung cancers. This however is related to the limited number of publications describing the radioprotector effects of polyphenols in breast tissue. We however feel these publications are relevant and therefore will keep these in the manuscript.
- Section 3 is definitely out of scope because between this and the earlier sections, authors cover every cancer!! Also, given the amount of literature that can potentially be covered in section 3, the write-up does not do justice.
Our review is focused on the effects of polyphenols on lung, breast and esophagus as most literature is related to these tissues. However, we think it is relevant to show that the radioprotective effects of polyphenols have also been described in other normal tissues. The mechanisms of polyphenols in these other tissues has been described to a lower extent, so as to keep the ideal length of the manuscript and keep our focus on lung, breast and esophagus cancer.
- Abstract is too vague and does not provide a snap shot of contents, for example, the polyphenols discussed, major mechanisms etc.
We agree with the reviewer, and modified the abstract accordingly, including a list of the major mechanisms of polyphenols (lines 24-25).
Reviewer 3 Report
The authors present a narrative review focusing on the protective properties of polyphenols against radiation-induced thoracic organ toxicities.
The title reflects the subject of the manuscript. It presents a clear and clinically useful message. It is well written in terms of clarity, style, and use of English language. The discussion section is sufficiently detailed and explains adequately the purpose of this study in the context of published information. The conclusion accurately and clearly explains the main result. The length of the manuscript is ideal. All references are appropriate and current.
Minor points:
Avoid using references in the conclusion section, these are concluding remarks.
Author Response
We greatly appreciate the positive comments of this reviewer. We have adjusted the conclusion section and removed references as suggested. See section 5 of the manuscript.
Reviewer 4 Report
I congratulate with the authors for their accurate, exhaustive overview on polyphenols either from the biochemical and application points of view.
I have not remarks.
Author Response
We would like to thank this reviewer for reviewing our manuscript; we greatly appreciate the positive feedback.
Reviewer 5 Report
The manuscript is novel and well written but I feel more figures should be there to explain the mechanism of action of polyphenols and even the chemical structures of the polyphenols should be there.
Author Response
We greatly appreciate the effort of this reviewer for the comments on our manuscript and positive feedback. We understand the concerns regarding the amount of figures. Therefore, we modified the first figure to make it more explanatory (see Figure 1), and we included a new figure (see Figure 2) with the chemical structures of the polyphenols as suggested.
Reviewer 6 Report
Radiotherapy, commonly used as a treatment for thoracic and breast cancers, has several adverse effects. Radioprotectors are compounds designed to reduce the damage in normal tissues caused by radiation. An ideal radioprotector should induce DNA repair mechanisms without radioprotective or toxicity effects on the tumour or other parts of the body. These compounds are often antioxidants and must be present before or at the time of radiation for their effectiveness. Several plant-derived products such as polyphenols have gained interest within oncology. This review focuses on summarizing the protective properties of polyphenols against radiation-induced thoracic organ toxicities. The manuscript reviews 205 articles regarding this topic. The topic of this manuscript is up‑to‑date, attractive and well-suited for your journal. The manuscript is well-written and divided into 5 main parts, the text is clear and easy to read. For better visualisation authors used 11 tables and 1 figure. I appreciate added abbreviation list at the end of this manuscript. I suggest checking for some spelling mistakes and grammar errors. Otherwise, I have no major concerns about this manuscript and I recommend it for publication.
Author Response
We would like to thank this reviewer for the positive comments as well as recommendation for publication. We have proofread again the manuscript and corrected spelling and grammar mistakes.
Round 2
Reviewer 2 Report
Authors have hardly accommodated any of my suggestions.
1. In response to my suggestion about reducing Introduction, authors declined to do as per my suggestion But I am OK with it.
2. About my concern on the inclusion of breast cancer, the response is not satisfactory. It would make sense to just focus on thoracic cancers and the protective effects on normal thoracic tissues. Inclusion of breast cancer dilutes the focus.
3. Similarly, authors also refused to part away with section 3 on 'other tissues'. If breast cancer and all other tissues are touched upon (albeit just barely), why does the title not simply mention all cancers. Authors need to understand that they provide a detailed account for just thoracic cancers and tissues and rest everything is out of focus and does not do justice considering the little information provided.
Author Response
- In response to my suggestion about reducing Introduction, authors declined to do as per my suggestion but I am OK with it.
We appreciate that the reviewer agreed with the current Introduction, which now excludes breast cancer from thoracic cancer as requested by the reviewer.
- About my concern on the inclusion of breast cancer, the response is not satisfactory. It would make sense to just focus on thoracic cancers and the protective effects on normal thoracic tissues. Inclusion of breast cancer dilutes the focus.
We removed all information on breast cancer and focused the main body of the review on thoracic cancers. Title has therefore also been changed accordingly.
- Similarly, authors also refused to part away with section 3 on 'other tissues'. If breast cancer and all other tissues are touched upon (albeit just barely), why does the title not simply mention all cancers. Authors need to understand that they provide a detailed account for just thoracic cancers and tissues and rest everything is out of focus and does not do justice considering the little information provided.
As indicated in point 2, we agree with the author that the information on breast cancer was inferior compared to other thoracic cancers and we therefore deleted this from the main body of the review. We adjusted the title accordingly, focusing it on thoracic tissues only. We however believe that beyond the main scope of this review (thoracic cancers), it is important to indicate to the reader that polyphenols do also have effects in other healthy tissues. Therefore, we would like to keep section 3 and related Table 11 as it currently is and hope that the reviewer will agree with this.
Round 3
Reviewer 2 Report
Authors have done the right thing by removing 'breast cancer'. They now need to take out the section on 'other tissues' as well because it serves no purpose. Also, it could be better utilized in some other follow-up publication from the group where the focus is more broad. Authors also need to understand that readers will not even visit this article looking for information on 'other tissues' when the scope is already defined in the title of the manuscript. Please do not diffuse the focus and scope of your article !!!
Author Response
We now moved the section on other normal tissues to the supplement and refer to this material in Discussion in case a reader becomes interested.
Round 4
Reviewer 2 Report
Thanks for addressing all of my concerns.